# Subunit interactions and arrangements in the fission yeast Mis16–Mis18–Mis19 complex

Melanie Korntner-Vetter[1],* , Stéphane Lefèvre[1],*, Xiao-Wen Hu[1], Roger George[2] , Martin R Singleton[1]

Centromeric chromatin in fission yeast is distinguished by the presence of nucleosomes containing the histone H3 variant Cnp1[CENP-A]. Cell cycle–specific deposition of Cnp1 requires the Mis16–Mis18–Mis19 complex, which is thought to direct recruitment of Scm3-chaperoned Cnp1/histone H4 dimers to DNA. Here, we present the structure of the essential Mis18 partner protein Mis19 and describe its interaction with Mis16, revealing a bipartite-binding site. We provide data on the stoichiometry and overall architecture of the complex and provide detailed insights into the Mis18–Mis19 interface.

## Introduction

Accurate segregation of eukaryotic chromosomes requires the formation of bipolar attachments between duplicated sister chromatids and the spindle poles. These attachments require specific loci to be marked on the chromosomes that act as sites for kinetochore formation. These loci, known as centromeres, must be capable of being propagated through DNA replication, mitosis, and transmitted to subsequent generations. In fission yeast and higher eukaryotes, the principal markers of centromeres are nucleosomes containing variant or posttranslationally modified histones rather than any particular underlying DNA sequence (Allshire & Karpen, 2008; McKinley & Cheeseman, 2016). Notably, all known centromeres are distinguished by the presence of a nucleosome containing the histone H3 variant, CENP-A (Cnp1 in *Schizosaccharomyces pombe*) (Earnshaw & Rothfield, 1985; Palmer et al, 1991; Takahashi et al, 2000). CENP-A–containing nucleosomes have the ability to bind inner kinetochore proteins, such as CENP-C and CENP-N, and so seed kinetochore formation (Okada et al, 2006; Foltz et al, 2006; Carroll et al, 2009). The mechanisms of CENP-A nucleosome loading and homeostasis remain incompletely understood. Extant CENP-A pools are diluted during DNA replication, and in fission yeast, they are replenished during the G2 phase of the cell cycle (Lando et al, 2012). A key player in the reloading process is the Mis18 complex, which seems to be the most upstream

factor required for deposition of CENP-A (Hayashi et al, 2004; Fujita et al, 2007). Mis18 is a coiled-coil protein containing an N-terminal YIPPEE domain, essential for centromere recruitment (Nardi et al, 2016; Subramanian et al, 2016; Stellfox et al, 2016). In humans, Mis18 recruitment to the centromere requires its association to Mis18BP1, a large SANT domain–containing protein, which can bind residual CENP-C from the previous cell cycle (Hayashi et al, 2004; Fujita et al, 2007; Maddox et al, 2007; Moree et al, 2011; Dambacher et al, 2012; Nardi et al, 2016). In fission yeast, Mis18BP1 is apparently not present; instead, Mis18 forms a complex with three smaller proteins: Mis16, Mis19 (also known as Eic1), and Mis20 (Eic2) (Hayashi et al, 2004, 2014; Hirai et al, 2014; Subramanian et al, 2014). Binding of Mis18 to Mis16 and Mis19 is a strict requirement for centromere localisation of the complex, whereas Mis20 seems dispensable. Mis16 is a homologue of the human WD40-repeat protein RbAp46/48, a well-known histone chaperone found in a variety of chromatin-associated complexes (Verreault et al, 1998; Parthun et al, 1996; Murzina et al, 2008), whereas Mis19 seems to form a bridge between Mis16 and Mis18 with its N terminus binding Mis18 and C terminus binding Mis16 (Hayashi et al, 2014; Subramanian et al, 2014) (Fig 1A). In addition to providing a mark for centromeric location, the Mis18 complex is thought to directly recruit Scm3-chaperoned Cnp1–H4 to the centromere (Pidoux et al, 2009; Williams et al, 2009; An et al, 2015). Current structural and biochemical data suggest a model in which Mis18 provides the principal link to the centromere, whereas Mis16 is responsible for recruitment of the Scm3–Cnp1–H4 complex, which allows Cnp1–H4 dimers to be inserted into chromatin as a prelude to complete nucleosome assembly. In this study, we provide new insights into the architecture of the Mis16–Mis18–Mis19 complex and discuss their implications for possible mechanisms of centromeric recruitment and histone deposition.

## Results

### Structure of Mis16 and interaction with histone H4

We first solved the crystal structure of *S. pombe* Mis16 both alone and with the N terminus (residues 14–44) of histone H4 bound at 1.9

[1]Structural Biology of Chromosome Segregation Laboratory, The Francis Crick Institute, London, UK   [2]Structural Biology Platform, The Francis Crick Institute, London, UK

Correspondence: martin.singleton@crick.ac.uk
*Melanie Korntner-Vetter and Stéphane Lefèvre contributed equally to this work.

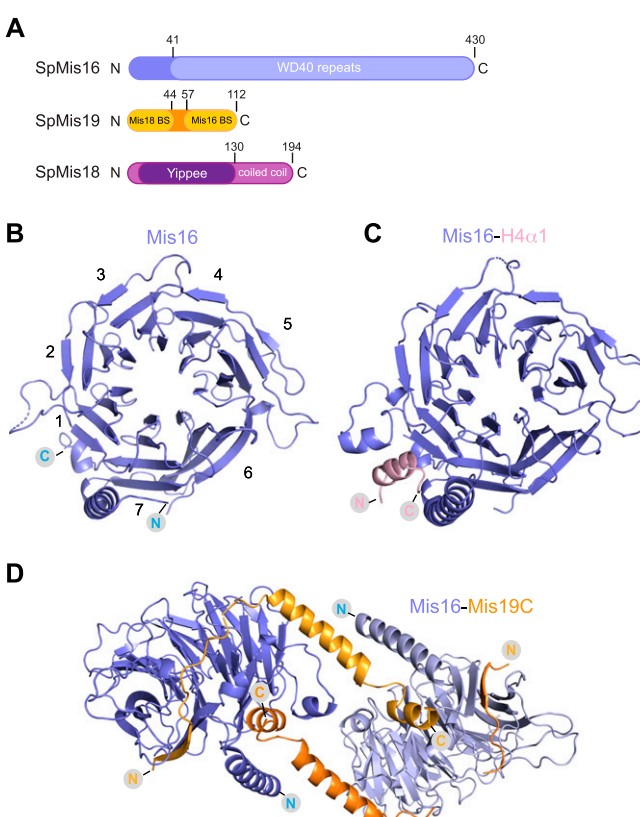

**Figure 1. Structural analysis of proteins.**
**(A)** Domain architecture of the *S. pombe* proteins Mis16, Mis19, and Mis18. Structural domains and putative binding sites are indicated. **(B)** Crystal structure of the *S. pombe* Mis16 protein. Propeller blades are numbered. **(C)** Crystal structure of Mis16 bound to the histone H4 peptide (residues 14–44, ordered residues 28–41). **(D)** Crystal structure of Mis16 protein bound to the residues 56–111 of Mis19 (orange). The dimer is formed by crystallographic twofold symmetry, and each Mis19 molecule bridges two Mis16 molecules.

and 1.8 Å resolution, respectively (Fig 1B and C and Table S1). As expected, Mis16 adopts a β-propeller fold, consisting of 7 WD-40 repeats, with an extended N-terminal α-helix that forms part of the histone interaction pocket (Fig 1B). The ordered N terminus of helix of histone H4 (H4α1, residues 28–41) sits in a concave pocket on the outside of the β-propeller fold formed by the Mis16 N terminus and a small insertion into blade 1 of the propeller (Fig 1C). The interaction is predominantly mediated by I35 and L38 of H4α1 forming a hydrophobic interface with Mis16 residues L40 and I416. Positively charged residues R40 and R41 of H4α1 interact with negatively charged residues E365, D366, and D369 of Mis16. Overall, the structure of *S. pombe* Mis16 and its interactions with histone H4 closely resemble those of the homologous *Schizosaccharomyces japonicus* Mis16 and the human RbAp46/48 proteins and exhibit a highly conserved interface (Fig S1A), which has been extensively described by others (Murzina et al, 2008; An et al, 2015).

## Structure of the Mis16–Mis19 complex

We identified a N-terminal truncation of Mis19 (residues 51–112, Mis19C) that was capable of interaction with Mis16. The crystal

structure of Mis16–Mis19C was solved at 2.0 Å by molecular replacement using Mis16 as a search model (Fig 1D). Mis19C forms an extended tether, which is able to bind two molecules of Mis16, one at each end. Residues 56–63 form a β-strand that continues β-propeller 7 of Mis16 and stabilises an extended loop from 107 to 116, which is disordered in the Mis16 structure alone (Fig 1B). Mis19 then loops across the C-terminal face of the Mis16 disc and then forms a helix (α1, residues 79–97) running diagonally across the edge of the H4-binding pocket. This helix leads into a short loop, and finally into the C-terminal helix (α2, residues 99–111), which sits in the H4-binding site of a symmetry-related Mis16 molecule. Analysis of the structure shows three main contacts between Mis16 and Mis19 (Fig 2A). The first (site A) is formed by the Mis19 β-strand and proximal loop. The second (site B) is formed by Mis19 residues 77–86 including a short loop region and the N-terminal part of α1. The third (site C) comprises the C-terminal helix that binds to the same binding pocket of Mis16 as H4α1 (Figs 1C and S2A).

## Mis16–Mis19 interactions

The C-terminal interaction of Mis19 with Mis16 (site C in our nomenclature) has been previously described (An et al, 2018). Our crystal structure reveals two additional binding sites between Mis16 and Mis19 (Figs 1D and 2A). To test the contribution of the individual sites to the Mis16–Mis19 interaction, various truncated Mis19 constructs were designed, for which site A, B, or both A and C are missing (Fig 2B). Full-length Mis16 and Mis19C (includes sites A, B, and C, residues 52–112), construct 1 (includes site B and C, residues 66–112), construct 2 (includes binding site B, residues 66–99), and construct 3 (includes site A and B, residues 52–99) were co-expressed in insect cells, as the isolated Mis19 truncations cannot be expressed and purified. The complexes were purified using a Strep-Tactin affinity tag on Mis16 and subsequent gel filtration step. All complexes elute from gel filtration as a single species. Corresponding SDS–PAGE gels showed that Mis19 is associated with Mis16 for all but construct 2, where only Mis16 could be detected (Figs 2C and S2B), suggesting that site B on its own is not sufficient to form a stable complex with Mis16. This site is formed by part of the unstructured region of Mis19 running across the top edge of Mis16. Given that the structure in this region is rather poorly ordered, and does not seem to be required for interaction as assessed by co-purification studies, we do not consider it further.

Interestingly, construct 3, which lacks binding site C, is still able to interact with Mis16, although rather more weakly than the full-length protein as judged by band intensities (Fig 2C). Taken together, the results suggest that the Mis19 C-terminal helix forms the main interaction with Mis16, but additional contributions are provided by site A. Further evidence that these sites are relevant in vivo comes from previously reported Mis19 temperature-sensitive (t.s.) mutants (Hirai et al, 2014; Subramanian et al, 2014). The A and C interaction sites are the location of the kis1-1 (R65C) and eic1-1 (F102S) mutations, respectively. Our structure allows us to rationalise the effects of these mutations (Figs 2D and S2C–E). The R65 side chain makes multiple salt bridges to two aspartates in Mis19 (D42 and D100), which would be lost in the cysteine mutant,

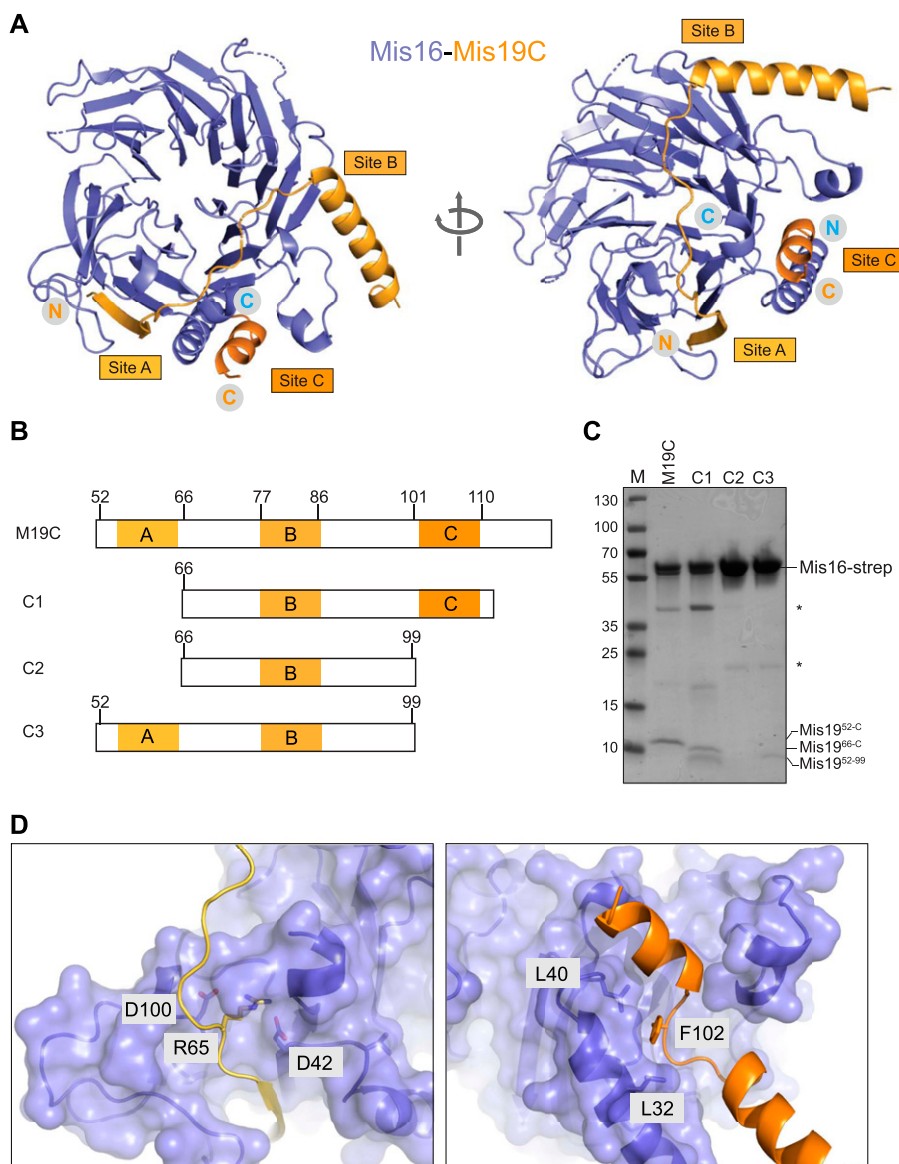

**Figure 2. Mis16–Mis19 contact regions.**
**(A)** Crystal structure of Mis16–Mis19 highlighting the Mis19-binding sites. Three putative binding sites A, B, and C (lighter to darker orange from N terminus to C terminus) are indicated. **(B)** Schematic of the various deletion constructs of Mis19 are used for binding studies showing the positions of sites A, B, and C. **(C)** Constructs containing affinity-tagged Mis16 and the Mis19 deletions shown above were co-expressed and purified by affinity pull-down and size-exclusion chromatography. SDS–PAGE gel of peak fractions is shown. Asterisks indicate impurities. **(D)** Close-up of the A- and C-binding sites. The locations of the kis1-1 (R65) and eic1-1 (F102) t.s. mutant residues are shown.

probably destabilising the interaction of the β1 strand with Mis16. In the case of the eic1-1 mutant, the F102 side chain stacks against the hydrophobic side wall of the binding pocket for the C-terminal helix on Mis16 and mutation to serine reduces but does not eliminate binding of Mis19 at this site (see below).

The A site is a previously unreported Mis16–Mis19 interaction. The Mis16 aspartates which Mis19 R65 bonds to are totally conserved in all Mis16 homologues including RbAp46/48, suggesting that this might be a conserved binding site for partner proteins (Fig S1B). Indeed, a very similar interaction between RbAp48[Rbbp4] and Suz12 in the polycomb repressor complex 2 (PRC2) complex has recently been reported (Chen et al, 2018). In this case, Suz12 forms an additional strand in the RbAp48 propeller, analogous to that seen in Mis19 (Fig S3A). Similar to Mis19, the interaction is mediated by a series of hydrophobic residues along the strand, capped by two basic residues that contact acidic side chains in the WD40 protein, although there is considerable sequence divergence between the proteins. Sequence alignments of Mis19 orthologs show that the A site is highly conserved with the critical Arg63 and Arg65 in particular being absolutely invariant (Fig S3B). Similarly, the Mis16 channel which the strand occupies also shows a high degree of conservation (Fig S3C), suggesting that this is a general binding site for partner proteins.

**Mis19–histone H4 competition**

Our structures of Mis16–H4α1 and Mis16–Mis19C show that the C-terminal α helix of Mis19C (Mis19α2) binds the same site on Mis16 as the histone H4α1 helix (Fig S2A). To better understand this dual specificity, we determined the affinities of the competing

interactions using isothermal titration calorimetry (ITC; Fig 3A). A peptide corresponding to histone H4α1 (residues 27–50) was found to bind to Mis16 with a $K_D$ of 29 nM, consistent with previous studies (An et al, 2018). Interestingly, Mis19α2 (residues 97–112) has considerably higher affinity, with a $K_D$ of 8 nM, a 3.6-fold increase in affinity over H4α1. To confirm that these are competitive interactions, we preincubated Mis16 with one peptide before determining the binding constant of the second (Fig 3B). The results showed that the Mis19α2 was capable of competing out histone H4α1 (apparent $K_D$ 2.5 μM), whereas the reverse experiment showed that H4α1 could not displace Mis19α2 at any concentration tested. The ITC experiments show that formation of the Mis16–Mis19 complex effectively prevents histone H4 binding to Mis16.

## Interaction of Mis16, Mis16–19, and the Mis16–18–19 complexes with full-length Cnp1–H4

To confirm that Mis19C binding prevents H4 interaction with Mis16 not only for the H4α1 construct but also for the full-length histone, we performed pull-down experiments using a histone H4–Cnp1 dimer (Figs 4A and S4A). We were unable to purify these directly as recombinant proteins, probably because of the lack of a suitable chaperone. Instead, we found that we could form Mis16–H4–Cnp1 by incubating purified Mis16 with Cnp1–H4–H2A–H2B complexes (histone octamers). We find that full-length Mis16 interacts with Cnp1–H4 and can displace H2A–H2B to form a Mis16–H4–Cnp1 complex in a pull-down assay (Fig 4A, pull-down lane 3). As free

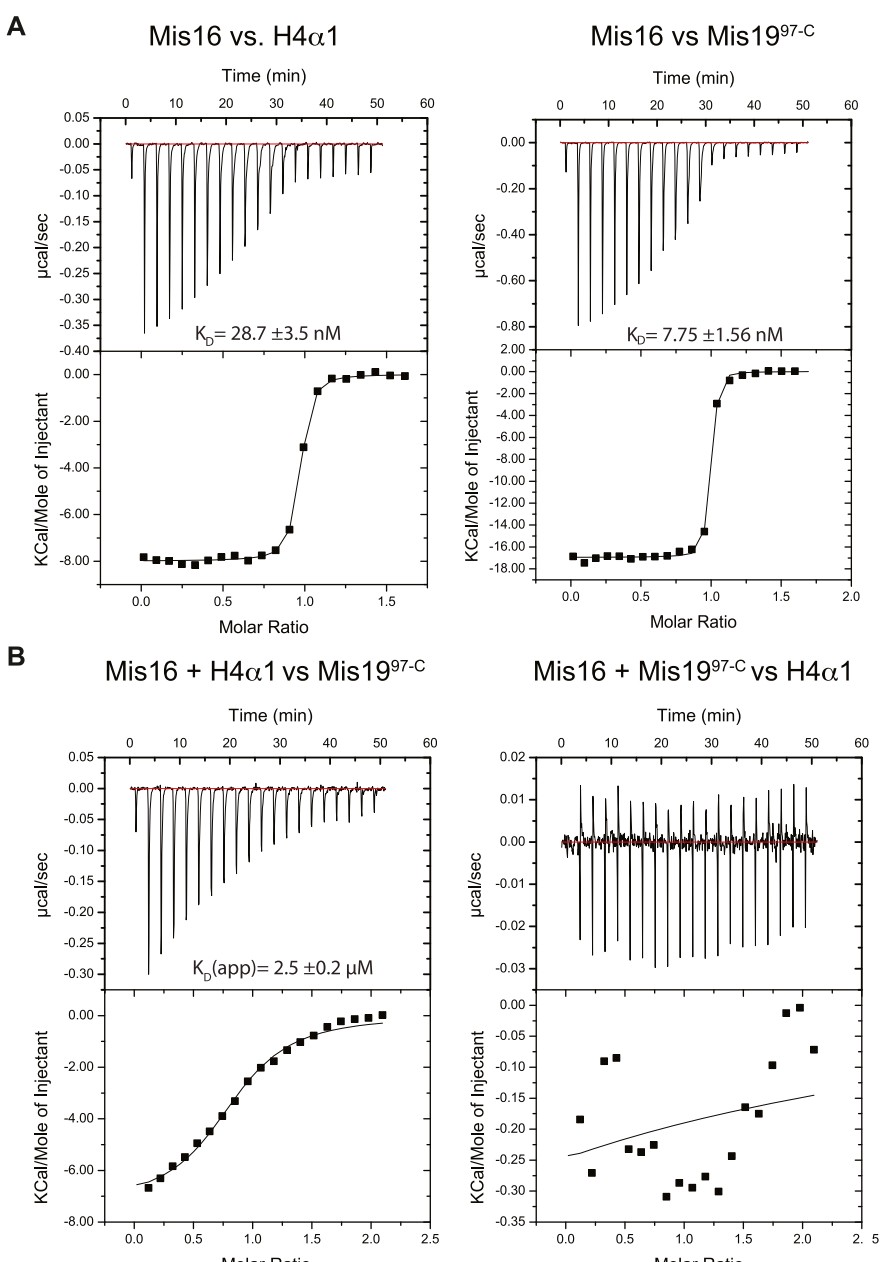

**Figure 3. Mis19 and histone H4 peptide–binding assays.**
**(A)** ITC measurements of binding affinity between Mis16 and H4α1 (left) or Mis19 site C (right) peptides. **(B)** ITC competition experiments. Mis16 was incubated with H4α1 (left) or Mis19 site C (right) peptides, before being challenged with the second substrate.

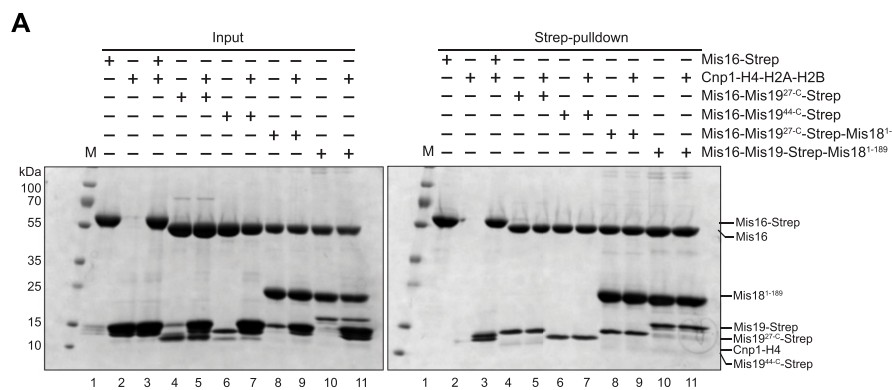

**A**

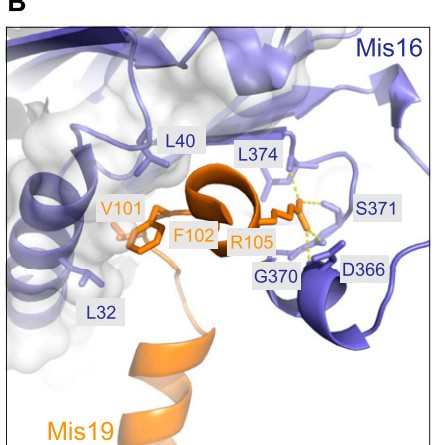

**B**

**C**

| Mutation | $K_D$ (nM) |
|---|---|
| wild-type | 8 |
| Mis19 V101S | 51 |
| Mis19 F102S | 301 |
| Mis19 R105A | > 1000 |
| Mis19 V101S F102S R105A | no binding |

**Figure 4. Mis19 and histone H4 competition and mutational analysis.**
**(A)** Pull-down assays to test competition between full-length histones Cnp1–H4 and Mis16–19 or Mis16–18–19 complexes. Left panel shows input SDS–PAGE gel of purified proteins and right panel shows results of pull-down assay. **(B)** Location of key residues in the Mis16–H4α1 interaction. **(C)** Binding constants of point and triple mutants of the Mis19 C-terminal peptide (residues 97–112) against Mis16, as determined by ITC.

Mis19 is extremely unstable, we tested if preformed Mis16–Mis19 dimers (full-length and N-terminally truncated, see sections below) could also interact with H4–Cnp1 by incubating them in a large molar excess of histone octamer. In all cases tested, preformed Mis16–Mis19 dimers, or the full Mis16–Mis18–Mis19 complex, showed identical stoichiometry (as judged by band intensity) regardless of the presence of histones, suggesting no displacement had occurred. We repeated the experiment using Strep-tagged Mis16 and untagged Mis19 and showed a similar result that the presence of Mis19 in the complex completely inhibits histone transfer to Mis16 (Fig S4B).

## Mutational analysis of Mis19 binding

To further understand the basis for Mis19 binding to Mis16 at site C, we analysed the contribution of three conserved residues in Mis19α2 that seem to predominantly mediate the interaction. V101 and F102 contribute to a hydrophobic interaction with the wall of the Mis16-binding pocket, whereas R105 forms hydrogen bonds to the side-chain hydroxyl of Mis16 S371 and backbone carbonyl group of Mis16 G370 (Fig 4B). Simultaneous mutation of all three residues to alanine or serine completely abolished the interaction as assessed by ITC (Figs 4C and S4C). Individual mutations in V101 or F102 reduced but did not abolish binding ($K_D$ ~ 51 and 301 nM, respectively, consistent with survival of the F102S mutant at

permissive temperature), whereas the R105A mutation substantially reduced binding ($K_D$ not determined).

## Oligomerisation of the Mis16–Mis19 complex

In our Mis16–Mis19C crystal structure, two Mis19 molecules bind two Mis16 molecules in trans, with sites A and B binding one Mis16, whereas site C binds the second molecule (Fig 1D). Given the extended and partially disordered nature of Mis19, it is conceivable that a single Mis19 molecule could bind a single Mis16 molecule with all three sites on Mis19, if site C were to "loop back" to contact Mis16. To further explore the stoichiometry of the Mis16–Mis19C complex, we performed solution mass measurements using multi-angle laser light scattering (MALLS) on the purified dimer (Fig 5A). The results showed a single population with a mass of 52 kD, probably corresponding to a Mis16:Mis19C dimer (calculated mass 55.8 kD). Such a dimer could be formed via the Mis19 A site, C site, or possibly both. Our pull-down experiments (Fig 2C) and ITC results would suggest that the A site is the highest affinity interaction, leading us to propose two possible models for the dimer arrangement (Fig 5C). We noticed that if the N-terminal affinity purification tag on Mis16 was not removed, we observed two peaks of 58 and 114 kD, corresponding to the Mis16:Mis19 dimer and a probable Mis16₂:Mis19₂ tetramer. It is possible that this tetramer is similar to the one observed in the crystal structure (Figs 1D and S5A and B). The tag is not present in the protein used for

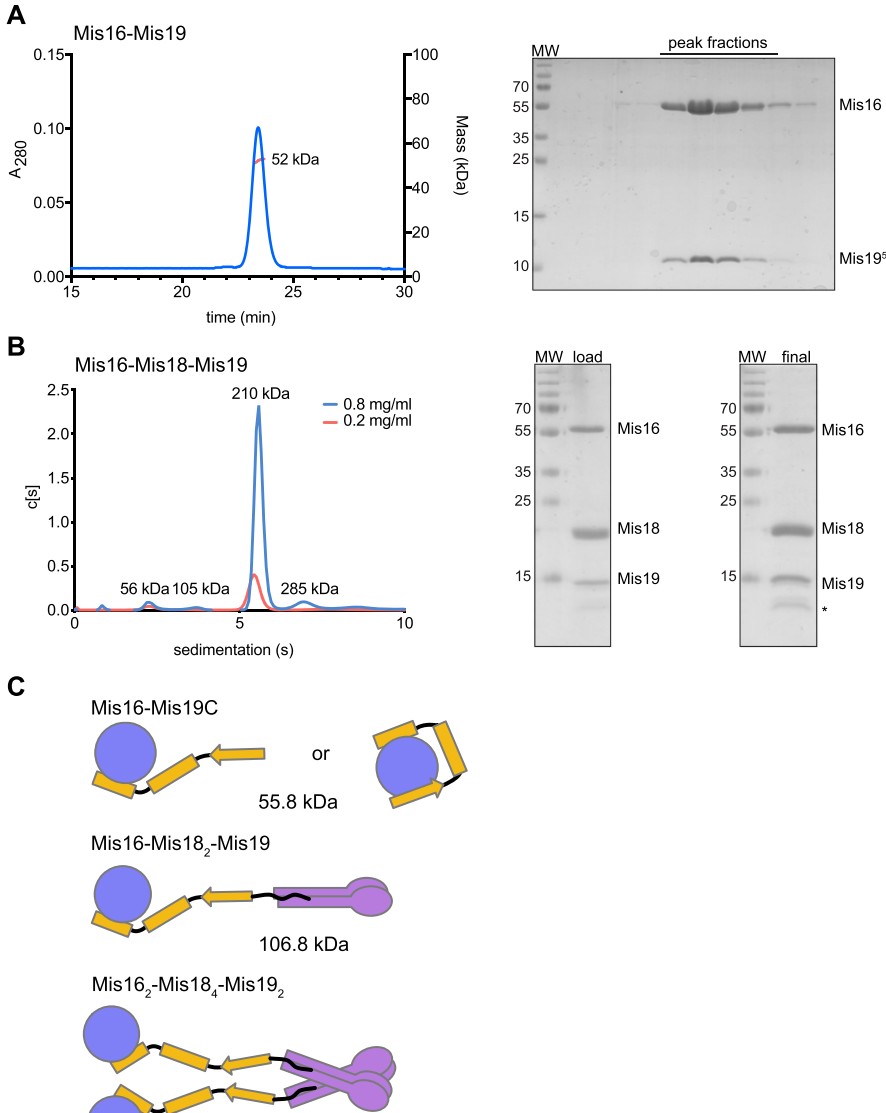

**Figure 5. Stoichiometry of the Mis18 complex.**
**(A)** MALLS measurement of mass of the Mis16–Mis19[52-C] complex. Right panel shows SDS–PAGE gel of elution fractions. **(B)** AUC analysis of the Mis16–Mis18–Mis19 complex at two concentrations (indicated). Right panel shows SDS–PAGE gels of input and recovered protein. **(C)** Schematic showing possible subunit arrangements that account for the observed solution masses. Different configurations are possible for the Mis16–Mis19 interaction depending on whether one or two binding sites are occupied, as indicated in the figure. Similar interactions could also occur in the larger complex but are omitted for clarity. Calculated masses are indicated for the untagged proteins.

the crystal structure (Fig S5C), and there is no obvious structural reason why it should be promoting tetramer formation in solution.

We next decided to further analyse the intact Mis16–Mis18–Mis19 complex. Full-length Mis16, Mis18, and Mis19 were simultaneously expressed in a baculovirus system and purified to homogeneity. During purification, we noticed that the complex appeared in differing oligomeric states as judged by size-exclusion chromatography. We determined the masses of the individual complexes using analytical ultracentrifugation (AUC) (Fig 5B). The dominant species has a mass of ~210 kD. Given that our previous results show that Mis16–Mis19 form a 1:1 complex, and Mis18 itself can dimerise via the YIPPEE domain and further tetramerise (Subramanian et al, 2016), the most likely arrangement for the ternary complex is that two Mis16–Mis19 dimers binds to a Mis18 tetramer via the N terminus of Mis19, giving an overall stoichiometry of Mis16₂–Mis18₄–Mis19₂. The smaller populations observed in the sedimentation distribution of 105 and 56 kD

most likely correspond to half the full complex, caused by dissociation of the Mis18 tetramer into dimers, and release of the Mis16–Mis19 complex from Mis18 (Fig 5C).

## Interactions between Mis19 and Mis18

It has previously been reported that the interaction between Mis18 and Mis 19 occurs via the N-terminal region of Mis19 binding to the C terminus (coiled-coil) of Mis18 (Hayashi et al, 2014). We tried to analyse this interaction in more detail. Given that residues 56–112 of Mis19 are involved interactions with Mis16, we constructed two truncations of Mis19 with half of the remaining N-terminal residues deleted in each case (Mis19[27-C] and Mis19[44-C]). We first asked if these Mis16–Mis19 constructs could still interact with Mis18 in an affinity pull-down experiment (Figs 6A and S6A). We confirmed that in the absence of Mis19, there is no direct interaction of Mis18 with Mis16

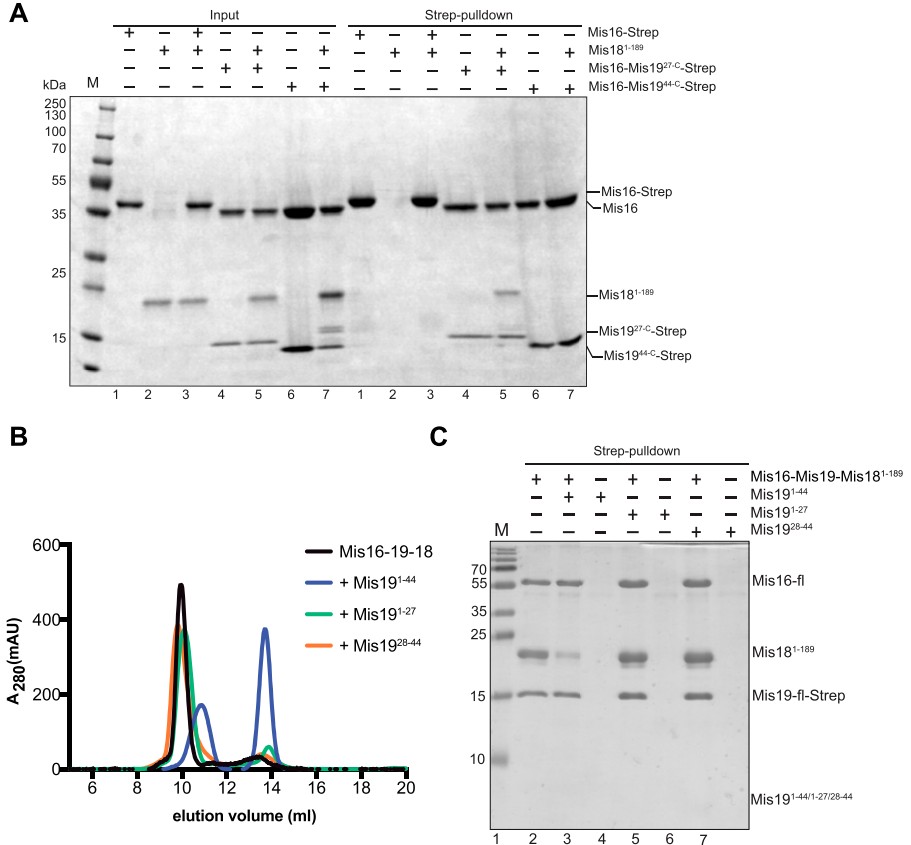

**Figure 6. The Mis19 N terminus binds Mis18.**
**(A)** Mis16–Mis18–Mis19 interactions assessed by pull-down. Input proteins are shown on the left. Binding of Mis16–Mis19 and Mis19 truncations to Mis18 are assessed by pull-down of Mis19 (right). **(B)** Mis16–19–18 binding assessed by size-exclusion and competition assays. Preformed Mis16–18–19 complex was challenged with peptides derived from the N terminus of Mis19 (indicated). Competitive interactions result in the dissociation of the complex into separate Mis16–19 and Mis18 peaks. **(C)** Pull-down assay of Mis16–Mis19–Strep–Mis18 complex in the absence (lane 1) and the presence of Mis19$^{1-44}$ (lane 2), Mis19$^{1-27}$ (lane 4), and Mis19$^{28-44}$ (lane 6) peptides.

(pull-down, lane 3). We then add Mis18 to the Mis16–Mis19 constructs. As expected, there is no interaction with the Mis19$^{44-112}$-containing complex (lane 7). Interestingly, however, we do see an interaction with the Mis19$^{27-112}$-containing construct (lane 5), although the Mis18 band seems somewhat weaker than in the full-length complex (compared with Fig 4A, lane 10), suggesting reduced affinity. To further analyse these interactions, we carried out competition assays, challenging preformed Mis16–Mis18–Mis19 complex with peptides from the N terminus of Mis19, analysing the results both by size-exclusion chromatography and affinity pull-down assays. As expected, introduction of Mis19$^{1-44}$ peptide effectively disrupts the complex, splitting it into a Mis18-containing and Mis16–Mis19–containing fractions (Fig 6B). This was confirmed by pull-down experiments (Fig 6C, lanes 1 and 2), although a small amount of more residual Mis18 was noticed, probably caused by the displacement not proceeding to completion. Introducing either the Mis19$^{1-27}$ or the Mis19$^{28-44}$ peptides had no effect on complex formation, consistent with both regions being required for robust binding. To try and pinpoint the residues that might be responsible for the interaction with Mis18, we carried out a sequence alignment of the Mis19 N terminus and noticed two conserved stretches of residues characterised by an IxxxF pattern (motif 1) and LxxF (motif 2) (Fig 7A). The two motifs are coincident with the two truncations we identified: between residues 1–27 and 28–44, respectively. As the N terminus of Mis19 is likely to be solvent-exposed (from our crystal structure), we reasoned that the conserved hydrophobic residues probably participate in protein–protein interactions. To test this, we

carried out another set of competition experiments using the intact Mis16–18–19 complex and peptides carrying mutations in motifs 1 and 2, separately or combined (Fig 7B). Consistent with our previous results, the wild-type peptide could effectively compete out Mis19 binding. Introduction of alanine substitutions in motif 1 or 2 either alone or in combination eliminated disruption of the complex formation as tested by size-exclusion (Fig 7C) or pull-down experiments (Fig 7D). Collectively, these results show that both motifs are important for mediating Mis19 binding to Mis18; however, deletion of motif 1 still seems to support a weaker interaction.

## Discussion

Here, we describe the architecture of the fission yeast Mis18 complex. The general arrangement involves Mis19 acting as a flexible linker between Mis16 and Mis18. We confirm previous observations that the N terminus of Mis19 binds the C terminus of Mis18, whereas the C terminus of Mis19 binds Mis16 (Hayashi et al, 2014; An et al, 2018). As previously described, the extreme C terminus of Mis19 binds to Mis16 by engaging the histone H4-binding pocket on Mis16. This interaction is of substantially higher affinity than the binding of H4 to Mis16 and in vitro at least, making the switch from H4 binding to Mis19 binding essentially irreversible. Previous studies have suggested a model in which Mis16 binds Scm3-chaperoned Cnp1–H4 dimers by recognition of both H4 and Scm3 (An et al, 2015). This model provides a nice explanation for the selectivity toward the centromeric histone. Our

**A**

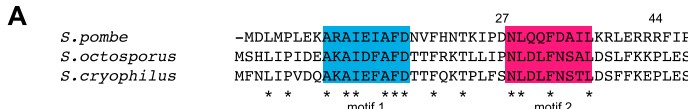

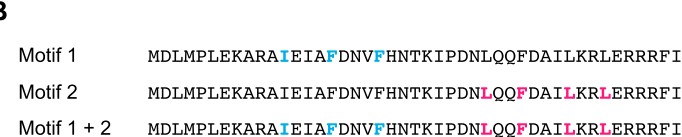

**Figure 7.  Details of the Mis19–Mis18 interaction.**
**(A)** Multiple sequence alignment of the N terminus of Mis19 orthologs. Two putative conserved regions were identified (blue and red). Asterisks indicate totally conserved residues. **(B)** Peptides used for competition assays containing conserved hydrophobic residues in the Mis19 N terminus. Residues in motif 1 (blue) or motif 2 (pink) were selected for alanine mutagenesis and competition binding analysis. **(C)** Repeat of competition assays in (6B) using peptides with alanine mutations in motif 1, motif 2 or both motifs. **(D)** Repeat of pull-down assays in (6C) using peptides with alanine mutations in motif 1, motif 2, or both motifs.

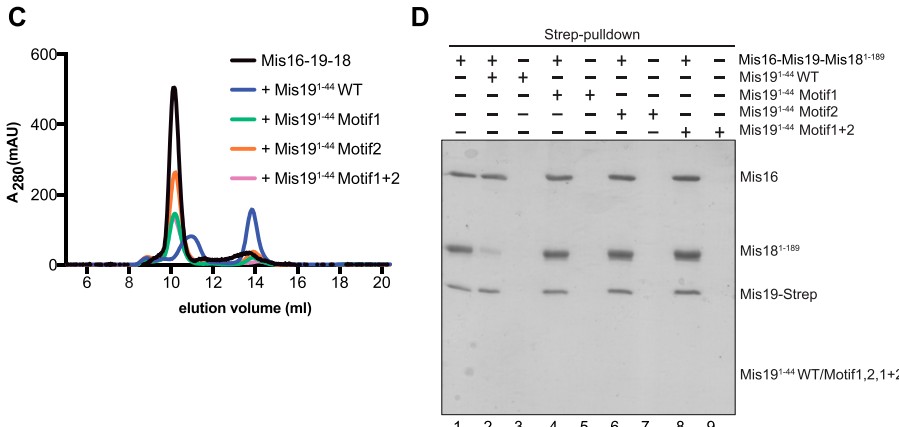

data presented here show that Mis16 can effectively bind a full-length Cnp1–H4 dimer in the absence of Scm3, supporting a model in which H4 recognition is independent of Scm3 interactions. Previous studies have demonstrated that the main interaction between histone H4 and WD40-type chaperones involves the H4 N-terminal helix (Verreault et al, 1998). Surprisingly, we also show that Mis16 can directly displace Cnp1/H4 from preformed histone complexes containing H2A/H2B. In intact nucleosomes, the H4 N terminus is presumed to be conformationally inaccessible for binding by Mis16 (Murzina et al, 2008), suggesting that the DNA-free histones are either conformationally more flexible and/or accessible to assembly factors or, alternatively, that additional interactions between Mis16 and the histones facilitate their disassembly. It would be interesting to see if the Cnp1/H4 displacement by Mis16 can also work with intact nucleosomes and whether this has any physiological role.

Our crystal structure, solved using a longer Mis19 construct than that previously studied, shows that there are two additional areas of contact between Mis16 and Mis19, which we call the "A" and "B" sites. The A site is created by the formation of a β-strand in Mis19 completing one propeller blade of the Mis16 structure. This mode of interaction may be common among the RbAp46/48 family and has been exploited in the context of the PRC2, where a so-called WDB2 domain in Suz12 forms a similar interaction with RbAp46. In this case, the interaction also seems to inhibit the binding of H3K4 to the complex (Chen et al, 2018). The A site seems to be a physiologically relevant interaction, given that it is the site of the kis1-1 t.s. mutant (Hirai et al, 2014). What might the function of this interaction be? One possibility is that it is required for the initial recruitment of Mis16 to centromeres. In the Mis16–H4–Scm3 trimer, the Mis16 H4-binding site

will be occupied by the N-terminal helix of the histone, and the complex presumably stabilised by additional indirect contacts through Scm3 (An et al, 2015). In this case, the effective affinity of the Mis19 C site for the complex would be low, but initial binding via the A site could occur, facilitating Cnp1–H4–Scm3 displacement by subsequent binding of the C site. This is consistent with the phenotypes of Mis19 mutants, where the kis1-1 mutant shows no localisation to centromeres, whereas the eic1-1 (site C) mutant shows reduced, but not eliminated, localisation (Hirai et al, 2014; Subramanian et al, 2014). We also note that the existence of a second Mis16-binding site on Mis19 explains previous studies, where an N-terminal truncation of Mis16 that removes L32, involved in mediating Mis19 binding via the C site, still shows substantial residual interaction (An et al, 2018). The same study shows that a slightly longer truncation, removing both Mis16 L32 and W33, exhibits considerably reduced binding. This can be explained by the observation that the W33 not only forms part of the hydrophobic cavity that binds the C site but also stabilises the β-propeller to which site A binds.

Our crystal structure shows that Mis16–Mis19C heterodimers can associate and form heterotetramers via reciprocal binding of the A and C sites in Mis19, but it is unclear if this tetramerisation occurs in solution under normal circumstances. Solution mass analysis shows a single species consistent with a $Mis16_1–Mis19_1$ stoichiometry. Binding studies suggest that both A and C sites can mediate Mis16–Mis19 interaction, as assessed by their individual deletions. In the case of a single Mis16–Mis19 heterodimer, it is conceivable that a single Mis19 molecule could bind a single Mis16 molecule using two sites on Mis19, if site C were to "loop back" to contact Mis16 (Fig 5C). In this case, the tetramer we see in the crystal

structure would simply be packing-induced, and the B site an artefactual consequence of this. An alternative possibility is that the crystallographic tetramerisation is physiologically relevant. Solution studies from us and others suggest that the stoichiometry of the full complex is Mis16$_2$–Mis19$_2$–Mis18$_4$, i.e., a dimer of Mis16–Mis19–Mis18$_2$ (An et al, 2018). It has been suggested that this dimerisation is mediated by Mis18, which has previously been shown to both dimerise and tetramerise independently of Mis16–Mis19 (Subramanian et al, 2016). It is possible that a second dimerisation interface could form in the full complex via the same interactions that we see in our crystal structure. However, this arrangement places the free Mis19 N-termini at opposite ends of the complex, which is hard to reconcile with the requirement for Mis18 tetramerisation without substantial structural rearrangements. Regardless of the exact arrangement, the presence of a Mis16 dimer in the complex seems obligate. Many lines of evidence have suggested that mature centromeric nucleosomes are octameric, containing two copies of CENP-A/H4 histone dimers (Camahort et al, 2009; Sekulic et al, 2010; Kingston et al, 2011; Tachiwana et al, 2011; Hasson et al, 2013; Nechemia-Arbely et al, 2017). If indeed Mis16 functions as Cnp1–H4 chaperone, it seems likely that a single Mis16 molecule binds a single Cnp1–H4 (or H3–H4) heterodimer, as demonstrated for the homologous RbAp48 protein (Zhang et al, 2012). In addition, the Scm3 protein seems to bind a single CENP-A/H4 dimer (Cho & Harrison, 2011; Hu et al, 2011; Zhou et al, 2011). In this case, two copies of the chaperoned histone dimer would be required for assembly of each centromeric nucleosome, which could be facilitated by Mis16–Mis18–Mis19 dimerisation.

Our experiments confirm that the binding of Mis19 to Mis18 requires the very N terminus of Mis19. In addition, we identify two conserved motifs in Mis19 that seem to mediate the binding. Each motif is characterised by a set of closely spaced, totally conserved hydrophobic residues. Both motifs seem necessary to mediate robust binding, although we still see a weaker interaction with Mis18 if motif 1 alone is deleted. Given that each Mis19 has to bind two Mis18 molecules, we speculate that each motif might be primarily responsible for binding a single Mis18 monomer, although further structural work will be required to confirm this.

# Materials and Methods

### Expression and purification of recombinant proteins

All proteins were expressed in insect cells using standard procedures. Genes were codon-optimised in all cases (GeneArt). Single protein constructs were cloned in pFastBac vectors (Invitrogen), whereas multi-protein complexes were cloned using the MultiBac system (Fitzgerald et al, 2006). The Mis16 protein carried a TEV protease-cleavable double-Strep-Tactin tag at the N terminus for purification purposes. All clones were sequence verified. Baculoviruses were generated and amplified using Sf21 cells, infected in Hi5 cells, and then further incubated at 27°C for 48 h. Harvested cells were lysed by sonication in buffer A containing 50 mM Tris–HCl, pH 7.5, 150 mM NaCl, 0.5 mM TCEP, benzonase nuclease (3 units/1 × 10$^6$

cells), 10% glycerol, and protease inhibitors cocktail set III (Calbiochem). After clarification by centrifugation, lysate was filtered through a 5-$\mu$m filter before chromatography. Proteins were purified using a 5-ml Strep-Tactin Superflow column (QIAGEN). Unbound proteins were washed out with lysis buffer, and the column was washed with buffer B (50 mM Tris–HCL, pH 7.5, 50 mM NaCl, and 0.5 mM TCEP) before elution with buffer B supplemented with 2.5 mM D-desthiobiotin. The eluate was applied onto a Poros HQ anion-exchange column (Applied Biosystems) and eluted with buffer C (50 mM Tris–HCl, pH 7.5, 1 M NaCl, and 0.5 mM TCEP). The N-terminal tag of Mis16 was cleaved off with TEV protease (4°C, overnight incubation). Protein was concentrated and applied to a Superdex S75 16/60 (Mis16) or S200 10/300GL (complexes) size-exclusion column (GE Healthcare) pre-equilibrated with buffer D (25 mM Tris–HCl, pH 7.5, 150 mM NaCl, and 1 mM TCEP). Fractions containing the complex were concentrated and either snap frozen in liquid nitrogen or used directly in crystallisation trials. Recombinant *S. pombe* centromeric histone octamers were formed by co-expression in *Escherichia coli* and purified using previously described procedures (Kingston et al, 2011).

### Crystallisation and structure solution

Crystals of all proteins were obtained using the sitting drop vapour diffusion method at 4°C. Crystallisation conditions for Mis16 were 0.1 M Na(OAc)$_2$, 0.8 M NaH$_2$PO$_4$, and 1.2 M KH$_2$PO$_4$, pH 4.5. Crystals of Mis16–H4$^{14–44}$ were grown in 0.05 M LiSO$_4$, 0.8 M NaH$_2$PO$_4$, 1.2 M K$_2$HPO$_4$, and 0.1 M CAPS, pH 9.5. Crystals of Mis16–Mis19C were grown in 0.6 M NaBr, 0.1 M Tris, pH 9.0, and 20% PEG 3350. Crystals were flash frozen and stored in liquid nitrogen. Data were collected under cryogenic conditions (100K) at Swiss Light Source beamline PXIII, ESRF beamline ID23, and Diamond beamline I04. Data sets were integrated and scaled using the Xia2-DIALS pipeline (Winter et al, 2018). The structure of Mis16 was solved by molecular replacement in Phaser (McCoy et al, 2007) using the human RbAp48 protein (PDB ID: 3CFS) as a search model. The refined Mis16 structure was subsequently used as a search model in molecular replacement solutions of the Mis16–H4 and Mis16–Mis19 structures. All structures were manually rebuilt using Coot (Emsley & Cowtan, 2004), with rounds of refinement in phenix.refine (Adams et al, 2010). Data collection and refinement statistics are presented in Table S1.

### AUC

The Mis16–Mis18–Mis19 complex was analysed in a buffer containing 150 mM NaCl, 25 mM Tris–HCl, pH 8.0, and 2 mM TCEP. All samples were concentrated to >0.1 mg/ml before loading into cells with identical buffer as a reference. Samples were run for 18 h at 50,000 rpm in an An50-Ti rotor (Beckman Coulter) at 4°C. Data were analysed using SEDFIT (Schuck, 2000), using a continuous c(s) distribution model with no prior knowledge. The partial specific volume of the protein was estimated as 0.734 cm$^3$/g.

### ITC

ITC experiments were performed at 30°C using an iTC200 (MicroCal). Samples were concentrated in a 50 mM Tris, pH 7.5, 100 mM NaCl, 4%

glycerol, and 0.5 TCEP buffer. Protein concentrations were determined using a spectrophotometer. The concentration of protein complex in the ITC cell was 26 $\mu$M. The lyophilised peptides of histone H4$^{27-50}$ or Mis19$^{97-112}$ were dissolved in the ITC buffer to a concentration of 260 $\mu$M. 3-$\mu$l injections were performed at 1,000 rpm stirring speed with an injection spacing of 3 min. The thermodynamic data were processed with the ORIGIN program (MicroCal).

## Multi-angle light scattering

200 $\mu$g of the Mis16–Mis19C complex was injected onto a Superose 6 Increase 10/300 column (GE Healthcare) coupled to a Wyatt Dawn 8+ MALS system. Data were analysed using Astra software.

## Pull-down assays

All in vitro pull-down assays were performed using purified proteins or protein complexes. A total of 30 $\mu$l of streptavidin–agarose beads (Thermo Fisher Scientific Pierce) per reaction were equilibrated in binding buffer containing 25 mM Tris–HCL, pH 7.5, 500 mM NaCl, 5% glycerol, and 1 mM TCEP. Strep-Tactin–tagged and untagged proteins were added to the equilibrated beads and incubated for at least 1 h at 4°C on a tube rotation wheel. Tagged bait proteins were at a concentration of 10–20 $\mu$M with untagged targets present in a twofold to fivefold molar excess. The beads were washed three times with 500 $\mu$l binding buffer and eluted from the beads with binding buffer supplemented by 25 mM desthiobiotin. Input and pull-down elution samples were analysed by SDS–PAGE.

## Data Availability

Atomic coordinates and structure factors for Mis16, Mis16–H4, and Mis16–Mis19C have been deposited in the Protein Data Bank under accession codes 6S1L, 6S1R, and 6S29, respectively.

## Supplementary Information

## Acknowledgements

We wish to thank the Crick Proteomics and Peptide Chemistry facilities for mass spectrometry band identification and supplying peptides, respectively. We also wish to thank the Crick Structural Biology Platform for assistance with ITC and MALLS studies, A. Purkiss for assistance with synchrotron data collection, and J. Wilson for helpful discussions about PRC structures. This work was supported by The Francis Crick Institute, which receives its core funding from Cancer Research UK (FC001155), the Medical Research Council (FC001155), and the Welcome Trust (FC001155).

## Author Contributions

M Korntner-Vetter: conceptualisation, data curation, formal analysis, investigation, visualisation, methodology, and writing—original draft, review, and editing.
S Lefevre: conceptualisation, data curation, formal analysis, investigation, visualisation, methodology, and writing—review and editing.
X-W Hu: investigation.
R George: data curation and investigation.
MR Singleton: conceptualisation, resources, data curation, formal analysis, supervision, funding acquisition, validation, methodology, project administration, and writing—original draft, review, and editing.

## Conflict of Interest Statement

The authors declare that they have no conflict of interest.

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
