## [Reviewer comments · Life Science Alliance]

Life Science Alliance

Subunit Interactions and Arrangements in the Fission Yeast Mis16-Mis18-Mis19 Complex

Martin Singleton, Melanie Korntner-Vetter, Stephane Lefevre, Xiao Wen Hu, and Roger George
DOI: <https://doi.org/10.26508/lsa.201900408>

Corresponding author(s): Martin Singleton,

Review Timeline:	Submission Date:	2019-04-26
	Editorial Decision:	2019-05-29
	Revision Received:	2019-07-16
	Editorial Decision:	2019-07-22
	Revision Received:	2019-07-23
	Accepted:	2019-07-23

Scientific Editor: Andrea Leibfried

Transaction Report:

May 29, 2019

Re: Life Science Alliance manuscript #LSA-2019-00408

Dr. Martin Robert Singleton
The Francis Crick Institute
Structural Biology of Chromosome Segregation Laboratory
The Francis Crick Institute
1 Midland Road
London NW1 1AT
United Kingdom

Dear Dr. Singleton,

Thank you for submitting your manuscript entitled "Subunit Interactions and Arrangements in the Fission Yeast Mis16-Mis18-Mis19 Complex" to Life Science Alliance. The manuscript was assessed by expert reviewers, whose comments are appended to this letter.

As you will see, the reviewers note that your work extends previous findings and they provide constructive input aiming at increasing the robustness of your results. We would thus like to invite you to submit a revised version of your work, addressing the concerns of the reviewers. We think doing so is straightforward and all concerns should get addressed, though it is not mandatory to address the second bullet point of rev#3 (site A mutational analysis).

Thank you for this interesting contribution to Life Science Alliance. We are looking forward to receiving your revised manuscript.

Sincerely,

B. MANUSCRIPT ORGANIZATION AND FORMATTING:

Reviewer #1 (Comments to the Authors (Required)):

The manuscript uses crystallography and biochemistry to resolve the interactions between the essential components of the yeast centromere assembly proteins. The complex of Mis16-Mis19-

Mis18 interacts with the centromere-specific histone chaperone Scm3 and directs the assembly of Cen1 nucleosomes to the centromere. Prior studies have identified sites of interaction between Mis16-Mis19, Mis19-Mis18, and Mis16-H4 histone. The work presented here confirms the results of these prior studies and identifies additional sites of interaction between Mis16 and Mis19. Moreover, it identifies important conserved residues that are essential for the interactions within Mis16-Mis19-Mis18 complex. In addition, the biochemical characterization using ITC, pull down assays, and analytical ultracentrifugation provide insights into binding specificity of different motifs within these proteins and hint at their stoichiometric relationship. These insights help towards developing a molecular model for centromere nucleosome assembly in yeast and might extend to other eukaryotes as some of these protein domains and interactions are conserved. Therefore, we support publication of this manuscript after the authors address the following concerns.

1. It is unclear if the results shown for copurification and streptactin-pull down experiments are representative of replicates or were only performed once. The pull down experiments need to be performed in replicates, since the authors make crucial conclusions about relative binding affinities etc. using this assay throughout the manuscript. In addition, the authors need to show the SDS-PAGE result for input for each of the binding reaction (and not just for the individual complexes or purified proteins as shown).

It is also unclear why the authors choose to perform Copurifications for some experiments and pulldowns using purified proteins for the others. The authors need to explain their rationale further.

2. The authors need to specify the molar concentration of individual proteins used in their pull down experiments.

3. Since the Streptactin tag appears to affect cis- vs. trans- interaction between the two binding sites of Mis19 on Mis16, the authors need to include SDS-PAGE of proteins used for Mis16-Mis19 crystallization to confirm lack of contamination from Mis16-Strep or Mis19-Strep (resulting from the incomplete protease cleavage of the Strep tag after affinity purification of these proteins).

4. In figure 3C, the authors claim to test if Mis19 (C site) binding blocks H4 (Cnp1-H4 nucleosome) binding to Mis16. The authors use Mis19-strep to pull down Mis16-Mis19 complex and find little to no nucleosome being pulled down with the complex. It is however unclear what fraction of the Mis16 is bound to nucleosomes in the presence and absence of Mis19 (without knowing the amount of Mis16, Mis19 and Cnp1 nucleosomes used in this experiment).

The author should consider confirming their results using Mis16-Strep as bait instead of Mis19-Strep to pull down the complex. If they see no nucleosome in their pull down, this would confirm that Mis19 binding outcompetes H4 binding. Additionally, since the authors have identified Mis19 (C site) mutations that appear to disrupt Mis16 binding, they should use this mutant in a reaction with Mis16-Strep and Cnp1 nucleosome to show that Cnp1 and not Mis19 mutant can be pulled down with Mis16.

5. The MALLS estimated mass does not agree completely with the theoretical mass calculated for the complexes shown in Figure 5. The reason for this discrepancy needs to be discussed. It is also unclear if the calculated masses are for full length, tagged, or truncated versions of the proteins in the complex.

6. Could Mis162: Mis191 be a possible explanation for the 114 kDa estimated mass from MALLS in supplemental figure S4 A?

7. It is unclear what the 285 kDa peak represents in the analytical ultracentrifugation in figure 5 B.

Reviewer #2 (Comments to the Authors (Required)):

This manuscript from the Singleton lab describes structural and/or biophysical characterization of the Mis16-Mis19-Mis18 complex from fission yeast that is required for propagation of centromeric

histone Cnp1/CENP-A at centromeres through the cell cycle. This work largely confirms studies on similar complexes from *S. pombe* and *S. japonicus*. Novel findings include structures of a larger fragment of Mis19, and identification of previously unidentified interactions with Mis16, demonstration that Mis19 can prevent Mis16-Histone H4 interactions in the context of a histone octamer, and further characterization of the N-terminal region of Mis19 with Mis18. The following points need to be addressed before publication.

Comments

1. Although the authors identify new interactions of Mis19 with Mis16, the structural models of the Mis16-Mis19 and Mis16-Mis19-Mis18 complexes are still unclear. Their in solution results would suggest that a single Mis16 molecule binds to one Mis19 molecule, which is at odds with the crystal structure. The authors seem to discount the importance of the "B" region of Mis19 due to their binding studies. If site B is truly unimportant, then are they suggesting that the contacts it makes with Mis16 in the structure are the results of a crystal packing artifact? If so, would this allow for the sites A and C to simultaneously engage Mis16? If site A is really exposed in the Mis16-Mis19 complex as diagrammed in Fig. 5C, shouldn't this result in dimers (possibly at higher concentrations)? All in all, the results appear to suggest that both sites can engage Mis16 simultaneously (if the data in Fig 5A can be confirmed, see below). The paper would be helped if the synthesis of these findings could be made more clear to the reader.

2. Regarding the biophysical characterization of the Mis16-Mis19 complex, the SDS-PAGE gel for the fractions in Fig. 5A should be displayed. Why is the experimental MW so much lower than expected for a complex? The MW of Mis16 on its own is 48.3 kDa, and the result in Fig 5A gives 52 kDa (vs 61.8 for a complex). Does this not suggest that the stoichiometry of this complex is not 1:1 under present conditions? The gel in Fig S4A would suggest this is a possibility. It is essential that the authors demonstrate that Mis19 is present in the peak in Fig. 5A.

3. In Fig. 3C, the authors use Cnp1 containing histone octamers to characterize the interactions with Mis16 +/- Mis19. Why is this substrate used and not a H4/Cnp1 dimer or a H4/Cnp1/Scm3 complex? Nonetheless, the eviction of H2B and H2A from the octamer by Mis16 is interesting. Do the authors believe this is due to the interaction with the H4 N-terminus alone or might other contacts be involved? If so, then site A may be playing a larger role in Mis16's chaperone role and Mis19's role in releasing Cnp1/H4. This should be discussed in the paper.

4. Another possible interpretation of the results in Fig S4 regarding the oligomerization state of Mis16-Mis19 is that the N-terminal tail of Mis19 may display weak dimerization, and it should be included as a possible configuration.

Minor point

The Cnp1 containing complex is not a "DNA-free nucleosome" as written in the text, it is a histone octamer.

Reviewer #3 (Comments to the Authors (Required)):

The manuscript by Korntner-Vetter and Lefèvre et al. aims at explaining subunit interactions and

arrangements in the fission yeast Mis16-Mis18-Mis19 complex. The authors report 3 crystal structures: spMis16, spMis16 in complex with N-terminus of H4 and complex between spMis16 and spMis19C. The first 2 structures provide limited novelty as analogous structures from yeast organisms were reported before. However, the structure of Mis16/Mis19C complex provides some new insights relevant for the field that the authors have further investigated using biophysical characterization. Mis16/Mis19C structure contains a longer Mis19 construct than the one used in study by An et al 2018. With the longer construct authors identify an additional relevant interaction site between 2 proteins (site A on Mis16). They dissect the contribution of particular aminoacids in site C by mutational analysis in combination with ITC and analyze oligomerization of Mis16/Mis19 and Mis16/19/18 by SEC-MALS and AUC. They identify 2 segments in N-terminus of Mis19 as important for interactions with Mis18.

Specific comments:

- The authors claim that they have identified another relevant interaction between Mis16 and Mis19 (site A). They should show electron density and distances for important interactions in this site and evolutionary conservation of the important interactions. Also, including pdb report from which one could assess the quality of the model in that region would be useful here.
- There is a comprehensive mutational study for the interactions in the site C that has been previously identified by others but similar study is missing for the site A that has been discovered in this study.
- For the interactions of the complexes with Cnp1, authors are using DNA-free nucleosomes. What is the physiological relevance for such an experiment? Wouldn't it be more appropriate to use Cnp1/H4 tetramer or even better Cnp1/H4/Scm3? Experiments with Cnp1 nucleosome (with DNA) would demonstrate if Mis16/19/18 can evict Cnp1/H4 from chromatin rather than working as a depositing factor.

Reviewer #1 (Comments to the Authors (Required)):

The manuscript uses crystallography and biochemistry to resolve the interactions between the essential components of the yeast centromere assembly proteins. The complex of Mis16-Mis19-Mis18 interacts with the centromere-specific histone chaperone Scm3 and directs the assembly of Cen1 nucleosomes to the centromere. Prior studies have identified sites of interaction between Mis16-Mis19, Mis19-Mis18, and Mis16-H4 histone. The work presented here confirms the results of these prior studies and identifies additional sites of interaction between Mis16 and Mis19. Moreover, it identifies important conserved residues that are essential for the interactions within Mis16-Mis19-Mis18 complex. In addition, the biochemical characterization using ITC, pull down assays, and analytical ultracentrifugation provide insights into binding specificity of different motifs within these proteins and hint at their stoichiometric relationship. These insights help towards developing a molecular model for centromere nucleosome assembly in yeast and might extend to other eukaryotes as some of these protein domains and interactions are conserved. Therefore, we support publication of this manuscript after the authors address the following concerns.

1. It is unclear if the results shown for copurification and streptactin-pull down experiments are representative of replicates or were only performed once. The pull down experiments need to be performed in replicates, since the authors make crucial conclusions about relative binding affinities etc. using this assay throughout the manuscript. In addition, the authors need to show the SDS-PAGE result for input for each of the binding reaction (and not just for the individual complexes or purified proteins as shown).

It is also unclear why the authors choose to perform Copurifications for some experiments and pulldowns using purified proteins for the others. The authors need to explain their rationale further.

We have repeated the pull-downs experiments as requested for figures 2C, 4A (previously Fig. 3C) and 6A and moved one copy to the supplementary information (Fig. S2B, S4A and S6A). When pull-down experiments are performed using purified proteins, we have included input samples of each binding reaction (Figures 4A and 6A). In the case of figure 6C and 7C, the competition Mis19 peptides cannot be clearly seen or illustrated on the SDS-Page gels because of their size, and so the input samples are not informative. Unfortunately, for the original pull-downs (Fig. S4A, S7A), input samples were not retained, but we did analyse the flow-through (unbound) fractions which confirm the excess of nucleosome in the input, and is now included in the figures. We would also like to add that in the case of Mis19/H4 competitive binding and Mis18-Mis19N binding (figures 4 and 6), our pull-downs are essentially confirming previous results, both from ourselves (e.g. ITC peptide binding assays) and others (mainly An et al., 2018). We therefore feel that the conclusions reached should be robust.

The rationale for performing pull-down experiments or co-purifications was dependent of whether it was possible to purify the isolated proteins. In the case of Mis19 in particular, neither the full-length nor truncated proteins are stable in isolation. Nevertheless, in order to carry out binding experiments, we purified the Mis19 fragments pre-bound to its partner Mis16. We have clarified in the text where and why we have done this.

2. The authors need to specify the molar concentration of individual proteins used in their pull down experiments.

We have aimed to use bait proteins at 10-20 μ M concentration, with target proteins present in at least a 2-5x molar excess. In some cases, concentration and yield of proteins (for example, isolated Mis18 in Figure 6A) affected how well we could approach this ideal. We have added the appropriate information to materials and methods section.

3. Since the Streptactin tag appears to affect cis- vs. trans- interaction between the two binding sites of Mis19 on Mis16, the authors need to include SDS-PAGE of proteins used for Mis16-Mis19 crystallization to confirm lack of contamination from Mis16-Strep or Mis19-Strep (resulting from the incomplete protease cleavage of the Strep tag after affinity purification of these proteins).

We have included a SDS-PAGE gel of the protein stocks used for crystallisation of Strep-tagged and untagged Mis16, showing that the tag is efficiently removed from the crystallised sample (figure S5C).

4. In figure 3C, the authors claim to test if Mis19 (C site) binding blocks H4 (Cnp1-H4 nucleosome) binding to Mis16. The authors use Mis19-strep to pull down Mis16-Mis19 complex and find little to no nucleosome being pulled down with the complex. It is however unclear what fraction of the Mis16 is bound to nucleosomes in the presence and absence of Mis19 (without knowing the amount of Mis16, Mis19 and Cnp1 nucleosomes used in this experiment).

We hope that the inclusion of the input gels for each binding reaction will allow a better judgment to be made as to the extent of binding (revised Figure 4A). The pull-downs were carried out in a ~5-fold molar excess of histone complex. Under these conditions we would expect that if any binding to Mis16 occurred, displacing Mis19, we would see an altered ratio of the Mis16-Mis19 band intensities in the pull-down. From the gels, the ratio appears

identical in both the presence and absence of histone complex. We agree that the experiment is not ideal in this case, but the main aim was to get qualitative answers to whether Mis19 competes against full-length H4, as an addition to our more quantitative peptide binding studies.

The author should consider confirming their results using Mis16-Strep as bait instead of Mis19-Strep to pull down the complex. If they see no nucleosome in their pull down, this would confirm that Mis19 binding outcompetes H4 binding. Additionally, since the authors have identified Mis19 (C site) mutations that appear to disrupt Mis16 binding, they should use this mutant in a reaction with Mis16-Strep and Cnp1 nucleosome to show that Cnp1 and not Mis19 mutant can be pulled down with Mis16.

We have repeated the pull-down experiments using Mis16-Strep as bait as suggested (figure S5B), and found that the presence of Mis19 52-C (the crystallised construct) blocks histone binding, in accordance with our previous results. The second experiment is not really possible to carry out, since we can only purify Mis19 when pre-bound to Mis16 (free Mis19 is probably rapidly proteolysed in the cell). Co-expression studies with a c-site deletion (e.g. figure 2C) show that Mis19 is weakly bound and rather rapidly degraded, resulting in a large excess of free Mis16. In this case, competition experiments against the Cnp1-H4 proteins would be uninformative.

5. The MALLS estimated mass does not agree completely with the theoretical mass calculated for the complexes shown in Figure 5. The reason for this discrepancy needs to be discussed. It is also unclear if the calculated masses are for full length, tagged, or truncated versions of the proteins in the complex.

We have clarified whether the proteins are tagged or not in the figure legends. The mass discrepancy was due to an error on our part. The calculated mass of 61.8 kDa was for a dimer of full-length Mis16-Mis19. The actual sample was full-length (untagged) Mis16 and Mis19 52-C (i.e. the same construct as used for crystallisation). This dimer has a predicted mass of 55.8 kDa, far closer to the MALLS value. We have included a gel to show that the Mis19 C-terminal truncation is indeed present in the sample (Figure 5A). This error also affected the calculations made for the tagged (probable tetrameric) species shown in figure S5, and we have updated the figure accordingly. We sincerely apologise for the confusion this has caused.

6. Could Mis16₂:Mis19₁ be a possible explanation for the 114 kDa estimated mass from MALLS in supplemental figure S4 A?

This is possible, but we feel that it is less likely than a Mis16(2)-Mis19(2) complex for two reasons. Firstly, the ratio of the band intensities on the SDS-PAGE gel are approximately similar for both the "dimer" and "tetramer" peaks, if anything the Mis16 band is slightly weaker in the larger peak, which would be inconsistent with a Mis16(2):Mis19(1) stoichiometry. Correcting for the error as described in the previous point would give a predicted mass for Mis16(2):Mis19 52-C(2) of 111.6 kDa, far closer to the observed value.

7. It is unclear what the 285 kDa peak represents in the analytical ultracentrifugation in figure 5 B.

We are not sure what this peak represents either. However, it has been previously shown that Mis18 can form higher order oligomers, so it is possible that this peak arises from this type of species.

Reviewer #2 (Comments to the Authors (Required)):

This manuscript from the Singleton lab describes structural and/or biophysical characterization of the Mis16-Mis19-Mis18 complex from fission yeast that is required for propagation of centromeric histone Cnp1/CENP-A at centromeres through the cell cycle. This work largely confirms studies on similar complexes from *S. pombe* and *S. japonicus*. Novel findings include structures of a larger fragment of Mis19, and identification of previously unidentified interactions with Mis16, demonstration that Mis19 can prevent Mis16-Histone H4 interactions in the context of a histone octamer, and further characterization of the N-terminal region of Mis19 with Mis18. The following points need to be addressed before publication.

Comments

1. Although the authors identify new interactions of Mis19 with Mis16, the structural models of the Mis16-Mis19 and Mis16-Mis19-Mis18 complexes are still unclear. Their in solution results would suggest that a single Mis16 molecule binds to one Mis19 molecule, which is at odds with the crystal structure. The authors seem to discount the importance of the "B" region of Mis19 due to their binding studies. If site B is truly unimportant, then are they suggesting that the contacts it makes with Mis16 in the structure are the results of a crystal packing artifact? If so, would this allow for the sites A and C to simultaneously engage Mis16? If site A is really exposed in the Mis16-Mis19 complex as diagrammed in Fig. 5C, shouldn't this result in dimers (possibly at higher concentrations)? All

in all, the results appear to suggest that both sites can engage Mis16 simultaneously (if the data in Fig 5A can be confirmed, see below). The paper would be helped if the synthesis of these findings could be made more clear to the reader.

We agree that the current presentation of these results is not as clear as it could be and have re-written the relevant section in the discussion to clarify our findings and potential models. To summarise these amendments and answer the reviewer's questions:

We discount site "B" on the basis of lack of direct binding to Mis16 (figure 2C) as assessed by pull-down and lack of obvious common structural features that would be expected to mediate such an interaction (we cannot of course exclude that it occurs). Our new results on the mass analysis of the Mis16-Mis19 complex (see next point) strongly suggest that a Mis16(1):Mis19(1) stoichiometry is most probable. In this case, the dimer of dimers observed in the crystal may result from crystal packing, making the "B" interaction an artefact as the reviewer suggests. In this case it is certainly plausible that both sites A and C could simultaneously engage a single molecule of Mis16 as the solution mass analysis indicates.

2. Regarding the biophysical characterization of the Mis16-Mis19 complex, the SDS-PAGE gel for the fractions in Fig. 5A should be displayed. Why is the experimental MW so much lower than expected for a complex? The MW of Mis16 on its own is 48.3 kDa, and the result in Fig 5A gives 52 kDa (vs 61.8 for a complex). Does this not suggest that the stoichiometry of this complex is not 1:1 under present conditions? The gel in Fig S4A would suggest this is a possibility. It is essential that the authors demonstrate that Mis19 is present in the peak in Fig. 5A.

This is due to an error on our part (please also see response 5 to reviewer 1). The calculated mass of 61.8 kDa was for full-length Mis16 and Mis19. The actual sample was full-length (untagged) Mis16 and Mis19 52-C (i.e. the same construct as used for crystallisation). This dimer has a predicted mass of 55.8 kDa, far closer to the MALLS value. We have included a gel to show that Mis19 truncation is indeed present in the sample (Figure 5A). We apologise for the confusion this has caused.

3. In Fig. 3C, the authors use Cnp1 containing histone octamers to characterize the interactions with Mis16 +/- Mis19. Why is this substrate used and not a H4/Cnp1 dimer or a H4/Cnp1/Scm3 complex? Nonetheless, the eviction of H2B and H2A from the octamer by Mis16 is interesting. Do the authors believe this is due to the interaction with the H4 N-terminus alone or might other contacts be involved? If so, then site A may be playing a larger role in Mis16's chaperone role and Mis19's role in releasing Cnp1/H4. This should be discussed in the paper.

We use histone octamers to form the complex as we were unable to express and purify H4-Cnp1 alone. This may be related to the lack of an appropriate chaperone, but we also find that *S. pombe* Scm3 is extremely difficult to handle. The histone octamer complex is relatively easy to work with, and allows us to prepare Mis16-Cnp1-H4 in a straightforward manner. Please see also response to point 3 from reviewer 3. From our data, it is hard to know whether release of H4-Cnp1 from the octamer is simply due to the N-terminal but this would be plausible, especially given the high affinity of the interaction. We have expanded our discussion on this point in the manuscript.

4. Another possible interpretation of the results in Fig S4 regarding the oligomerization state of Mis16-Mis19 is that the N-terminal tail of Mis19 may display weak dimerization, and it should be included as a possible configuration.

We have included a model showing this configuration in figure S5B.

Minor point

The Cnp1 containing complex is not a "DNA-free nucleosome" as written in the text, it is a histone octamer.

We have amended the text to reflect this.

Reviewer #3 (Comments to the Authors (Required)):

The manuscript by Korntner-Vetter and Lefèvre et al. aims at explaining subunit interactions and arrangements in the fission yeast Mis16-Mis18-Mis19 complex. The authors report 3 crystal structures: spMis16, spMis16 in complex with N-terminus of H4 and complex between spMis16 and spMis19C. The first 2 structures provide limited novelty as analogous structures from yeast organisms were reported before. However, the structure of Mis16/Mis19C complex provides some new insights relevant for the field that the authors have further investigated using biophysical characterization. Mis16/Mis19C structure contains a longer Mis19 construct than

the one used in study by An et al. 2018. With the longer construct authors identify an additional relevant interaction site between 2 proteins (site A on Mis16). They dissect the contribution of particular aminoacids in site C by mutational analysis in combination with ITC and analyze oligmerization of Mis16/Mis19 and Mis16/19/18 by SEC-MALS and AUC. They identify 2 segments in N-terminus of Mis19 as important for interactions with Mis18.

Specific comments:

- The authors claim that they have identified another relevant interaction between Mis16 and Mis19 (site A). They should show electron density and distances for important interactions in this site and evolutionary conservation of the important interactions. Also, including pdb report from which one could asses the quality of the model in that region would be useful here.

We have included a new figure (figure S2C) showing the electron density for the site A interaction with Mis16, and included more details on the key interactions (figures S2D and S2E). We have also included new figures showing the surface conservation of Mis16 in this region (figure S3C) and a sequence alignment of the C-terminal Mis19 binding sites A, B and C (Figure S3B). We have expanded our discussion of the evolutionary conservation in the main text and provided a PDB validation report for the structure for review purposes, showing that the model is of high quality in the interacting region.

- There is a comprehensive mutational study for the interactions in the site C that has been previously identified by others but similar study is missing for the site A that has been discovered in this study.

We agree that such an analysis would be useful, but feel that it would be beyond the scope of this study. In addition, we note that a mutant corresponding to the main specific interaction in the C-site (Mis19 Arg65, kis1-1) has been previously analysed by Hirai et al. (2014).

- For the interactions of the complexes with Cnp1, authors are using DNA-free nucleosomes. What is the physiological relevance for such an experiment? Wouldn't it be more appropriate to use Cnp1/H4 tetramer or even better Cnp1/H4/Scm3? Experiments with Cnp1 nucleosome (with DNA) would demonstrate if Mis16/19/18 can evict Cnp1/H4 from chromatin rather than working as a depositing factor.

The experiment has limited physiological relevance. The original aim of the experiment was to see if (free) Mis19 could eject Cnp1/H4/Scm3 from Mis16 as some models have suggested. Unfortunately, we could not produce Cnp1/H4 or Cnp1/H4/Scm3 in useable quantity, and free Mis19 alone is also not stable. We found that we could generate Mis16/Cnp1/H4 by incubating Mis16 with histone octamers, so we decide to test whether Mis19 could compete against this transfer using pre-formed Mis16-Mis19 complexes. We agree that the experiment is not as strong as we would like, but the results were fully in accordance with our peptide binding studies and results from An et al., so we felt it was worth including the results in the manuscript. We have amended the manuscript text to make this clearer. It would be very interesting to see if the complex could remove Cnp/H4 from genuine nucleosomes as the reviewer suggests, but we feel that such work is outside the scope of the current study.

July 22, 2019

RE: Life Science Alliance Manuscript #LSA-2019-00408R

Dear Dr. Singleton,

Thank you for submitting your revised manuscript entitled "Subunit Interactions and Arrangements in the Fission Yeast Mis16-Mis18-Mis19 Complex". As you will see, the reviewers appreciate the introduced changes and we would thus be happy to publish your paper in Life Science Alliance pending final revisions necessary to meet our formatting guidelines.

- please address the minor comment of rev#3
- please upload all figures (also suppl figures) as individual files and the suppl table in docx or xls format; the suppl figure legends can get added to the main manuscript text
- please add a callout to figure S1A in the manuscript text

A. FINAL FILES:

B. MANUSCRIPT ORGANIZATION AND FORMATTING:

Full guidelines are available on our Instructions for Authors page, <http://www.life-science->

alliance.org/authors

Sincerely,

Andrea Leibfried, PhD
Executive Editor
Life Science Alliance
Meyerohofstr. 1
69117 Heidelberg, Germany
t +49 6221 8891 502
e a.leibfried@life-science-alliance.org
www.life-science-alliance.org

Reviewer #1 (Comments to the Authors (Required)):

The authors have addressed my major concerns with the manuscript and it is now acceptable for publication.

Reviewer #2 (Comments to the Authors (Required)):

The authors have addressed my concerns and have improved the paper with new data and changes to the text. I recommend this manuscript for publication in LSA.

Reviewer #3 (Comments to the Authors (Required)):

Authors have addressed my comments and provided additional structural analysis and electron density map for relevant site. They have also modified the discussion, explaining their experiments in a more clear way. I support publishing of their study.

Minor comment: Figure S2E that is showing electrostatic interactions in site A should be also colour coded by atoms (not only by chains).

July 23, 2019

RE: Life Science Alliance Manuscript #LSA-2019-00408RR

Dear Dr. Singleton,

Thank you for submitting your Research Article entitled "Subunit Interactions and Arrangements in the Fission Yeast Mis16-Mis18-Mis19 Complex". It is a pleasure to let you know that your manuscript is now accepted for publication in Life Science Alliance. Congratulations on this interesting work.

DISTRIBUTION OF MATERIALS:

Again, congratulations on a very nice paper. I hope you found the review process to be constructive and are pleased with how the manuscript was handled editorially. We look forward to future exciting submissions from your lab.

Sincerely,

Andrea Leibfried, PhD
Executive Editor
Life Science Alliance
Meyerhofstr. 1

69117 Heidelberg, Germany
t +49 6221 8891 502
e a.leibfried@life-science-alliance.org
www.life-science-alliance.org